# Collaborative Action of Microglia and Astrocytes Mediates Neutrophil Recruitment to the CNS to Defend against *Escherichia coli* K1 Infection

**DOI:** 10.3390/ijms23126540

**Published:** 2022-06-11

**Authors:** Peng Liu, Xinyue Wang, Qian Yang, Xiaolin Yan, Yu Fan, Si Zhang, Yi Wei, Min Huang, Lingyan Jiang, Lu Feng

**Affiliations:** 1The Key Laboratory of Molecular Microbiology and Technology, Ministry of Education, Nankai University, Tianjin 300457, China; 1120170078@mail.nankai.edu.cn (P.L.); 1120200566@mail.nankai.edu.cn (X.W.); yang1125qian@163.com (Q.Y.); 2120211278@mail.nankai.edu.cn (X.Y.); 1120170077@mail.nankai.edu.cn (Y.F.); 1120170076@mail.nankai.edu.cn (S.Z.); 1120170081@mail.nankai.edu.cn (Y.W.); huangmin0304@163.com (M.H.); 2Tianjin Key Laboratory of Microbial Functional Genomics, TEDA Institute of Biological Sciences and Biotechnology, Nankai University, Tianjin 300457, China

**Keywords:** *E. coli* K1, outer membrane vesicle, microglia, astrocyte, neutrophil

## Abstract

*Escherichia coli* K1 is a leading cause of neonatal bacterial meningitis. Recruitment of neutrophils to the central nervous system (CNS) via local immune response plays a critical role in defense against *E. coli* K1 infection; however, the mechanism underlying this recruitment remains unclear. In this study, we report that microglia and astrocytes are activated in response to stimulation by *E. coli* K1 and/or *E. coli* K1-derived outer membrane vesicles (OMVs) and work collaboratively to drive neutrophil recruitment to the CNS. Microglial activation results in the release of the pro-inflammatory cytokine TNF-α, which activates astrocytes, resulting in the production of CXCL1, a chemokine critical for recruiting neutrophils. Mice lacking either microglia or TNF-α exhibit impaired production of CXCL1, impaired neutrophil recruitment, and an increased CNS bacterial burden. C-X-C chemokine receptor 2 (CXCR2)-expressing neutrophils primarily respond to CXCL1 released by astrocytes. This study provides further insights into how immune responses drive neutrophil recruitment to the brain to combat *E. coli* K1 infection. In addition, we show that direct recognition of *E. coli* K1 by microglia is prevented by the K1 capsule. This study also reveals that OMVs are sufficient to induce microglial activation.

## 1. Introduction

The morbidity and mortality rates associated with bacterial meningitis remain high despite advances in antimicrobial therapy [1]. *Escherichia coli* K1, a Gram-negative bacterium, is among the leading causes of meningitis in neonates and young infants, with mortality rates of infected neonates being as high as 40% and up to half of the survivors suffering permanent neurological sequelae [2,3]. Bacterial meningitis usually develops through the survival/multiplication of *E. coli* K1 in the intravascular space, which results in bacteremia. Bacteria then traverse the blood–brain barrier (BBB), surviving and multiplying in the subarachnoid space [4,5], which induces an inflammatory response with pro-inflammatory cytokine and chemokine production, resulting in early leukocyte recruitment [6,7,8]. Neutrophils are typically the first leukocytes to be recruited to an inflammatory site and are capable of eliminating pathogens through multiple mechanisms, including generation of reactive oxygen species (ROS), release of protease-enriched granules, and formation of neutrophil-extracellular traps (NETs) [9]. Neutrophils have been shown to be essential for the host to control *E. coli* K1 infection, as depletion of mouse neutrophils resulted in higher bacterial loads in the cerebellum and increased mortality [10]. However, the mechanisms associated with neutrophil recruitment to the central nervous system (CNS) to control *E. coli* K1 infection are not fully understood.

In general, recruitment of neutrophils to the site of inflammation or infection is a complex process involving interactions between neutrophils, resident immune cells, and several other resident cells—such as glial and endothelial cells in the brain—through various adhesion molecules and chemokines [11]. During infection, recognition of pathogens by resident immune cells leads to the release of pro-inflammatory cytokines, such as TNF-α and IL-1β, which activate endothelial cells to express adhesion molecules (such as ICAM-1) and several other resident cell populations (e.g., glial and endothelial cells) to express chemokines (such as CXCL1 and CXCL2), guiding neutrophil migration [12,13]. Neutrophils harbor a variety of chemokine receptors, including CXC family chemokine receptors CXCR1 and CXCR2, which can respond to chemokines and then migrate to infection foci [14]. The binding of integrins expressed on the surface of neutrophils and endothelial adhesion molecules triggers signals that lead to the opening of endothelial junctions, followed by neutrophil extravasation via paracellular and transcellular routes [15].

Inflammatory response in the CNS plays a critical role in neutrophil recruitment and defense against bacterial infections. Activation of microglia and astrocytes is a prominent feature of CNS inflammatory response [16]. Microglia are resident immune cells of the CNS and act as the first sentinels to protect against invading pathogens crossing the BBB [17]. Microglia are activated upon detection of pathogen-associated molecular patterns (PAMPs) by cell-surface-localized pattern recognition receptors (PRRs), such as toll-like receptors (TLRs) [17]. TLR activation induces subsequent downstream signaling events, resulting in the expression and release of antimicrobial pro-inflammatory cytokines, such as TNF-α and IL-1β [18]. Astrocytes are the most abundant cell type in the CNS and are also known to exhibit immune functions in the CNS [19]. Astrocytes can be activated by pro-inflammatory cytokines secreted from activated microglia, and these activated astrocytes can release various cytokines and chemokines to amplify neuroinflammation, thus combating infection [16]. At present, it is unclear whether the activation of microglia and/or astrocytes contributes to host defense against *E. coli* K1 infection and whether their activation is required for the recruitment of neutrophils into the CNS.

Gram-negative bacteria constitutively secrete outer membrane vesicles (OMVs, ~20–250 nm in diameter) into the extracellular milieu [20]. OMVs play an important role in the activation and modulation of the host’s immunity. First, OMVs possess the ability to activate the host’s innate immune system because they carry a variety of bacterial PAMPs, including lipopolysaccharides (LPS), membrane proteins, and nucleic acids, which can be recognized by PRRs located at the cell surface of immune cells [21]. Second, OMVs have been identified as vehicles that mediates the cytosolic localization of PAMPs during extracellular Gram-negative bacterial infections to trigger cytosolic inflammasome assembly and activation [22]. Moreover, OMVs have been found to mediate the delivery of virulence factors into host cells, including immune cells, to modulate host immunity and promote pathogenesis [21,23]. However, the interactions between *E.-coli*-K1-secreted OMVs and the immune response in the CNS have not been studied.

In this study, we investigated the mechanisms underlying neutrophil recruitment to the CNS during *E. coli* K1 infection of the brain. We found that microglia respond to *E.-coli*-K1-secreted OMVs and release the pro-inflammatory cytokine TNF-α, which then activates astrocytes, leading to the release of the chemokine CXCL1 by astrocytes. CXCL1 promotes recruitment of CXCR2-expressing neutrophils to the CNS. Together, our data revealed a complex immune defense mechanism in response to *E. coli* K1 infection of the brain that involves the collaborative action of microglia and astrocytes, providing new insights into CNS antibacterial immunity.

## 2. Results

### 2.1. E. coli K1 Induces Neutrophil Recruitment into CNS

To investigate the dynamic interactions between neutrophil recruitment and bacterial growth in the brain after *E. coli* K1 invasion into the CNS, we intravenously injected 3-week-old mice with *E. coli* K1 strain RS218 or PBS (negative control). The number of neutrophils and bacterial load in the mouse brain were assessed 4, 12, 24, and 48 h post-infection (hpi). Neutrophils were detected in the brain as early as 4 hpi and reached a peak at 24 hpi but decreased substantially at 48 hpi, and the neutrophils infiltrate mainly the meninges (Figure 1a and Appendix A). In line with the neutrophil levels, the bacterial load in the infected mouse brain was also detected at 4 hpi, reached a peak at 24 hpi, and then declined rapidly (Figure 1b). Neither neutrophil infiltration nor bacteria were detected in the brains of the mice injected with PBS (Figure 1a,b). These results indicate that neutrophils respond quickly to *E. coli* K1 infection in the CNS and are then recruited to control bacterial growth in the brain.

To verify the contribution of neutrophil recruitment to the control of *E. coli* K1 infection, mice were infected with *E. coli* K1 for 4 h to induce meningitis and thereafter treated intraperitoneally with either Ly-6G antibody, to deplete mouse neutrophils from circulation, or an isotype control [24,25]. The number of neutrophils and bacterial load in the mouse brain were assessed 24 hpi. The number of neutrophils in the brains of anti-Ly-6G-antibody-treated mice was significantly lower compared to that in the brains of isotype-treated mice (Figure 1c), while the bacterial load in the brains of anti-Ly-6G-antibody-treated mice was significantly higher (Figure 1d), indicating that neutrophil depletion impeded control of bacterial replication in the CNS. Collectively, these data suggest that the CNS initiates antibacterial immunity against *E. coli* K1 infection via neutrophil recruitment.

### 2.2. E.-coli-K1-Induced Neutrophil Recruitment Requires Microglia-Derived PRO-Inflammatory Cytokine TNF-α

The pro-inflammatory cytokines TNF-α and/or IL-1β have been reported to trigger the recruitment of neutrophils [26,27]. To assess whether these two cytokines are involved in neutrophil recruitment during *E. coli* K1 CNS infection, we first investigated the levels of TNF-α and IL-1β in the brains of infected mice. *E.-coli*-K1-infected and control mice were euthanized 4, 12, 24, and 48 hpi. Mouse brain tissue was used for RNA extraction, and the mRNA levels of TNF-α and IL-1β were analyzed using real-time quantitative PCR (qPCR). The results showed that the mRNA levels of both TNF-α and IL-1β were increased in the brains of K1-infected mice at all time points tested and reached their peak 12 hpi (Figure 2a). As cytokines are mostly secreted into the extracellular matrix to mediate immune response, we then performed enzyme-linked immunosorbent assay (ELISA) to measure the protein levels of TNF-α and IL-1β in the extracellular supernatant derived from *E.-coli*-K1-infected mouse brains. The results showed that TNF-α protein levels increased from 4 hpi and remained high 12 and 24 hpi, while IL-1β protein was not detectable at any of the time points tested (Figure 2b). The observed discrepancy between IL-1β mRNA and protein levels may be explained by the lack of mature IL-1β secreted into the extracellular matrix due to the fact that IL-1β is expressed and synthesized in microglia as an inactive precursor protein that requires cleavage by caspase-1 to be transformed into its mature form and because only the mature IL-1β can be secreted into extracellular space [28]. Thus, it is possible that TNF-α is involved in the recruitment of neutrophils to the CNS during *E. coli* K1 infection. To further test whether TNF-α is responsible for neutrophil recruitment to the CNS, we infected wild-type mice and TNF-α-deficient mice with *E. coli* K1 for 24 h and then quantified neutrophils in the brains of infected mice. We found that the number of neutrophils was significantly lower in TNF-α-deficient mice than in wild-type mice after *E. coli* K1 challenge (Figure 2c), indicating that TNF-α is required for *E.-coli*-K1-induced neutrophil recruitment.

TNF-α can be released by microglia and astrocytes in the CNS [29,30]. We next investigated which cell type is responsible for the increased TNF-α production in *E.-coli*-K1-infected mouse brains. As microglia are the first responders in the CNS, we first tested whether they contribute to TNF-α production during *E. coli* K1 infection of the CNS. We treated mice with the CSF-1R antagonist (PLX3397) for two weeks to deplete their microglia [31], and vehicle (10% DMSO and 90% corn oil) administration was used as a negative control. PLX3397-treated and control mice were then infected with *E. coli* K1, and the production of TNF-α in the brain was assessed by ELISA. The results showed that the production of TNF-α was significantly decreased in the brains of PLX3397-treated mice compared to that in control mice (Figure 2d,e), indicating that microglia contribute to the increased TNF-α production in *E.-coli*-K1-infected mouse brains. The decrease in TNF-α production in PLX3397-treated mice also resulted in significantly decreased neutrophil recruitment and an increased bacterial load in the brain (Figure 2f,g). As astrocytes can also produce TNF-α, we compared the TNF-α levels produced by microglia and astrocytes. Primary microglia and astrocytes were isolated from the brains of mice and infected with *E. coli* K1. The cellular supernatants were collected at the indicated time points, and TNF-α protein levels were determined by ELISA. The results showed that although astrocytes can produce TNF-α when stimulated with *E. coli* K1, the TNF-α levels produced by microglia were higher than those produced by astrocytes, indicating that TNF-α is mostly produced by microglia in the CNS (Figure 2h). Taken together, our data indicate that *E. coli* K1 stimulates TNF-α release by microglia, which drives the recruitment of neutrophils into the CNS.

### 2.3. Microglia Recognize E.-coli-K1-Derived OMV to Release TNF-α

We next attempted to identify the pathogen-associated factors that induce microglial TNF-α production. As resident macrophages of the CNS, microglia recognize invading pathogens via PRRs and engulf them to eliminate pathogens and activate inflammatory responses. Interestingly, we found that only a small portion of microglia (approximately 10%) contained engulfed bacteria when culturing microglia with *E. coli* K1 despite the high multiplicity of infection (MOI) (10 bacteria per cell) used (Appendix A). Given that a large proportion of *E. coli* K1 resists direct recognition by microglia, it is surprising that microglia are activated to induce immune responses. Gram-negative bacteria can secrete OMVs into the extracellular milieu, which carry diverse PAMPs [21]. Moreover, it has been reported that before invasion of the CNS, OMVs released by *E. coli* K1 can promote bacterial interaction with the bone marrow microvascular endothelial cells [32]. Therefore, it is possible that microglia recognize *E.-coli*-K1-derived OMVs, thereby triggering release of the pro-inflammatory cytokine TNF-α. To verify this hypothesis, we first confirmed the production of OMVs upon culturing microglia with *E. coli* K1. Transmission electron microscopy (TEM) and confocal laser microscopy revealed that a large number of OMVs were secreted by *E. coli* K1 (Figure 3a,b). Next, we analyzed whether *E.-coli*-K1-derived OMVs activated microglia. We cultured *E. coli* K1 and microglia in a Transwell system (Figure 3c), with the BV2 microglia cell line cultured in the lower wells and *E. coli* K1 loaded in the lower or upper wells to stimulate BV2 cells for 2 h. Cellular supernatant in the lower well was collected, and ELISA was used to assess TNF-α production. The results showed that *E. coli* K1 in the upper wells clearly drove robust production of TNF-α, although to a slightly lesser degree than that induced by bacteria in the lower wells (Figure 3d), indicating that microglia not only interact with *E. coli* K1 directly, but also sense factors secreted by the bacteria to mediate inflammatory response. In addition, we purified *E.-coli*-K1-derived OMVs to stimulate microglia and measured TNF-α production. We found that the OMV-induced TNF-α production by microglia was equivalent to that produced by microglia cultured with bacterial supernatant containing all bacteria-secreted factors (Figure 3e). To further verify the role of OMVs in stimulating TNF-α production during K1 infection, we generated hypovesiculating Δ*ypjA* and hypervesiculating Δ*yrbE E. coli* K1 mutant strains, which have reduced (Δ*ypjA*) and increased (Δ*yrbE*) abilities to produce OMVs, respectively. We used the mutants to stimulate microglia and measured the resulting TNF-α levels. The results showed that the production of TNF-α by microglia following stimulation with the mutants was significantly decreased for Δ*ypjA* and increased for Δ*yrbE* compared to the wild-type strain (Figure 3f). Together, these data indicate that *E.-coli*-K1-derived OMVs are the major elements that activate microglia and induce TNF-α production.

*E. coli* K1 may resist microglial phagocytosis because the presence of the K1 capsule enhances anti-phagocytic properties of *E. coli* K1. To verify this hypothesis, we constructed a *neuB*–*neuD* double mutant strain (Δ*neuDB*) that is deficient in K1 capsule production. We found that Δ*neuDB* was more easily engulfed by microglia than the wild-type strain when cultured with BV2 cells (Appendix A), implying that the K1 capsule helps the bacteria resist phagocytosis and avoid direct recognition by microglia.

### 2.4. Microglial TLR4 Recognizes LPS from E. coli K1-Derived OMVs to Release TNF-α

OMVs contain various PAMPs, including LPS, lipoproteins, DNA, and RNA, which can be recognized by microglia via PRRs. TLRs are the most commonly reported PRRs for sensing OMVs and activating immune responses [21]. To determine which TLRs are responsible for recognizing *E. coli* K1-derived OMVs, primary microglia were treated with either dimethyl sulfoxide (DMSO) as a control or with inhibitors of TLR2 (C29), TLR3 (CU CPT 4a), TLR4 (Resatorvid), TLR5 (TH1020), and TLR8 (CU-CPT9b) before or after incubation with OMVs. ELISA was used to assess TNF-α levels. The results showed that treatment of microglia with C29, CU CPT 4a, and CU-CPT9b had no effect on the release of TNF-α, but treatment of microglia with Resatorvid and TH1020 caused a significant decrease in TNF-α production (Figure 4a), indicating that TLR4 and/or TLR5, but not TLR2, TLR3, or TLR8, was required for the production of TNF-α. Considering that TH1020 not only inhibits TLR5 activity but also directly represses the expression of the TNF-α signaling pathway, we investigated whether TLR5 is involved in the activation of microglia when stimulated with OMVs using primary microglia derived from *TLR4*- and *TLR5*-deficient mice. After stimulation with OMVs, the cellular supernatant of TLR4^−/−^ or TLR5^−/−^ microglia was collected, and ELISA was used to assess TNF-α levels. We found that the TNF-α levels produced by the TLR4^−/−^ microglia were significantly decreased compared to those of the wild-type microglia, but the TNF-α levels produced by TLR5^−/−^ microglia were comparable to those of the wild-type microglia (Figure 4b), suggesting that TLR4, but not TLR5, is required for sensing OMVs and inducing TNF-α production. In addition, we generated a mutant strain Δ*msbB*, which has an impaired lipid A structure due to the lack of the myristic acid moiety of lipid A and thus cannot stimulate TLR4 response. OMVs derived from the mutant and K1 wild-types were used to stimulate microglia. We observed that when stimulated with OMVs derived from Δ*msbB,* microglia produced significantly decreased TNF-α levels compared to those stimulated with OMVs derived from wild-type *E. coli* K1 (Figure 4c), indicating that LPS is required to induce production of TNF-α by OMVs. In addition, after *E. coli* K1 challenge, mice lacking TLR4 but not TLR5 exhibited significantly reduced TNF-α production and neutrophil recruitment in comparison with wild-type mice (Figure 4d–g), indicating that recognition of LPS by TLR4 was required for TNF-α production and neutrophil recruitment.

We further attempted to determine which signaling pathways downstream of TLR4 are associated with *E.-coli*-K1-OMV-induced microglial TNF-α release using inhibitors of various signaling pathways. We found that BAY11-7082 (NF-κB inhibitor), but not T-5224 (c-Fos/AP-1 inhibitor) or SB203580 (p38 MAPK inhibitor), completely abrogated OMV-induced TNF-α release from microglia (Figure 4h), indicating that NF-κB is involved in TLR4 pathway-dependent TNF-α release from OMV-activated microglia. Together, these data indicate that OMV-induced production of TNF-α by microglia is dependent on the TLR4-NF-κB signaling pathway.

### 2.5. Astrocyte-Derived Chemokine CXCL1 Is Required for Neutrophil Recruitment to CNS during E. coli K1 Infection

Innate immune response is usually primed by pro-inflammatory cytokines; however, neutrophil recruitment requires chemotactic cues and chemokine recognition by the chemokine receptors of neutrophils. CXC chemokine receptor 2 (CXCR2) is a prominent chemokine receptor on neutrophils that has been implicated in TNF-α-induced neutrophil recruitment [33,34]. To investigate whether CXCR2 plays a role in neutrophil recruitment in *E.-coli*-K1-infected brains, we analyzed neutrophil migration induced by the cellular supernatant derived from whole *E.-coli*-K1-infected brain tissue via in vitro transmigration assays [35]. Primary neutrophils isolated from the bone marrow of 6-week-old mice were treated with either CXCR2 antibodies or an isotype control before inducing neutrophil transmigration, and transmigrated cells were counted after 3 h. The results showed that treatment of neutrophils with CXCR2 antibody significantly decreased neutrophil transmigration, with the magnitude of change being less (approximately 50%) than that in the isotype-treated control (Figure 5a). Similar results were observed in CXCR2-inhibitor-treated mice, since injection of AZD-5069, a CXCR2 inhibitor, led to >60% inhibition of neutrophil infiltration in *E.-coli*-K1-infected brains compared to the vehicle-treated mice (Figure 5b). These results demonstrate that CXCR2 is required for neutrophil recruitment to the *E.-coli*-K1-infected mouse brain. CXCR2 is the receptor for several ELR+ chemokines, including two potent neutrophil chemoattractants, CXCL1 and CXCL2 [36]. To determine whether CXCL1 and/or CXCL2 promoted neutrophil recruitment in *E.-coli*-K1-induced meningitis, mice were subjected to intracranial co-injection of *E. coli* K1 with a blocking antibody against CXCL1 or CXCL2, and neutrophil recruitment was analyzed using FACS. The results showed that anti-CXCL1-treated mice exhibited significantly reduced neutrophils in the brain compared to those found in the isotype-treated control mice (Figure 5c), and anti-CXCL2 did not cause obvious inhibition compared to the isotype antibody (Figure 5d). Our data are in agreement with a previous report that CXCL1 plays an important role in neutrophil recruitment in fungus-infected brains [37]. It should be emphasized that during neutrophil transmigration towards inflamed tissues, CXCL2 plays an important role in unidirectional paracellular neutrophils but is mostly produced by neutrophils [34]. These data indicate that the CXCL1–CXCR2 chemokine axis is critical for neutrophil recruitment.

Bacterial meningitis has been reported to induce robust CXCL1 expression in microglia and astrocytes [29,30,38,39]. We further investigated which glia were responsible for the production of CXCL1 by infecting mice with *E. coli* K1 and then assessing CXCL1 expression in the brain. Following *E. coli* K1 infection, whole brains were cryosectioned using a microtome and then observed under a confocal microscope. The results showed that CXCL1 expression did not colocalize with microglia (Iba1^+^ cells) but colocalized with astrocytes (GFAP^+^ cells) (Figure 5e), indicating that CXCL1 is mainly produced by astrocytes. Together, these data indicate that astrocyte-derived chemokine CXCL1 is indispensable for neutrophil recruitment in *E.-coli*-K1-infected brains.

### 2.6. TNF-α Produced from Microglia Promotes CXCL1 Release by Astrocytes

Microglia are constant sensors of changes in the CNS microenvironment. They serve as primary immune cells of the CNS and regulate the innate immune functions of astrocytes via cytokines. Previous studies have reported that astrocytes respond to TNF-α treatment with a typical transition to polygonal morphology and a shift to an inflammatory phenotype [40]. TNF-α also induces an NF-κB-dependent CXCL1 increase in cultured astrocytes [41]. Here, we hypothesized that microglia-secreted TNF-α promotes the release of CXCL1 from astrocytes in the *E.-coli*-K1-infected brain. To test this hypothesis, PLX3397-treated mice and TNF-α-deficient mice were infected with *E. coli* K1 for 12 h, and the production of CXCL1 in the brain was analyzed by ELISA. We found that the production of CXCL1 was markedly reduced in PLX3397-treated mice and TNF-α-deficient mice compared to that in the respective controls (Figure 6a,b), indicating that microglia-derived TNF-α plays an important role in the production of CXCL1 in the *E.-coli*-K1- infected brain. This result was further confirmed in vitro. After OMV stimulation, primary microglia-conditioned medium (MCM) or TNF-α-deficient microglia-conditioned medium (TNF-α^−/−^ MCM) was collected, and CXCL1 production from astrocytes in response to MCM or TNF-α^−/−^ MCM was detected. The results showed that the production of CXCL1 in astrocytes induced by TNF-α^−/−^ MCM was significantly decreased compared to that in MCM-treated astrocytes (Figure 6c). In addition, when primary astrocytes were stimulated with TNF-α for 2 h in vitro, TNF-α induced the production of CXCL1 by astrocytes in a concentration-dependent manner (Figure 6d). Moreover, when primary astrocytes were pretreated with the TNF-α receptor antagonist R-7050 and then treated with TNF-α, CXCL1 levels were significantly reduced compared to that in the vehicle-treated control (Figure 6e). Together, these data indicate that TNF-α produced by microglia promotes astrocytic CXCL1 production.

## 3. Discussion

Neutrophils are essential for the early control of CNS *E. coli* K1 infections [10]. However, the mechanisms by which neutrophils are recruited from circulation into the CNS to defend against *E. coli* K1 are unclear. Here, we provide key mechanistic insights into neutrophil recruitment to the CNS during *E. coli* K1 infection. We report that microglia and astrocytes work together in the CNS to recruit neutrophils, with microglia responding first to *E. coli* K1 infection to release the pro-inflammatory cytokine TNF-α, which activates astrocytes to release the chemokine CXCL1, which triggers the migration of neutrophils. Depletion of either microglia-and/or astrocyte-derived CXCL1 elicits weak neutrophil recruitment and increased bacterial load in the CNS. We also show that CXCR2-expressing neutrophils primarily respond to CXCL1 released by astrocytes. This study provides a new paradigm for the collaborative action of different cell types inside the CNS to control bacterial infection.

As resident innate immune cells in the CNS, microglia are considered immune sentinels of the brain and play a key role in the pathophysiology of bacterial meningitis [29]. Previous studies have shown that once pathogens gain access to the brain after crossing the BBB, microglia can recognize bacterial products such as LPS via PRRs, resulting in the phosphorylation of NF-κB, which plays a major role in initiating the inflammatory cascade [42]. Although it is known that prestimulation of microglia can increase phagocytosis and killing of *E. coli* K1 [43,44,45], the role and associated mechanisms of microglial cells in resisting *E. coli* K1 brain infection by recruiting immune cells in vivo have not been characterized. Our work clearly confirmed that microglia contribute to the host’s defense against *E. coli* K1 infection in vivo by initiating neutrophil recruitment. We showed that microglia, via TLR4, recognize LPS of *E. coli* K1 and then secrete TNF-α to initiate the recruitment of neutrophils to the CNS. Depletion of either microglia or microglia-derived TNF-α prevents recruitment of sufficient neutrophils and impairs bacterial clearance in the brain. Moreover, we showed that production of TNF-α by microglia is also dependent on the NF-κB signaling pathway. Thus, our study also provides a paradigm of microglial activation in response to bacterial infection via TLR4 and the NF-κB signaling pathway.

Notably, we found that microglia are activated primarily by *E.-coli*-K1-derived OMVs and microglia recognize *E. coli* K1 LPS carried by OMVs via TLR4 to release TNF-α. We also showed that the presence of the K1 capsule helps bacteria resist phagocytosis and avoid direct recognition by microglia. It has been reported that increasing the uptake of *E. coli* K1 by microglia can enhance intracellular killing of the bacteria [43,44]; hence, employing a K1 capsule to resist phagocytosis by microglia would be beneficial to the survival of the bacteria both in vitro and in vivo. Thus, our results reflect the arms race between the host’s CNS immune system and *E. coli* K1, with the K1 capsule mediating bacterial resistance to host immunity, while host immune cells have the ability to recognize bacteria-derived OMVs to initiate host immune functions. OMVs are known to be constitutively produced by all types of Gram-negative bacteria in a variety of environments and enable bacteria to communicate with their environment. Recognition of OMVs by microglia, leading to the release of pro-inflammatory cytokine TNF-α, undoubtedly contributes to the host defense against K1 infection. Moreover, it is known that the production of OMVs increases when bacteria are exposed to stressful conditions such as the host milieu [46]. Thus, it seems surprising that K1 produces increased levels of OMVs that can be recognized by immune cells to defend against infection without benefiting its own survival. In addition to carrying PAMPs such as LPS, OMVs are also known to carry virulence factors of pathogens and mediate the delivery of virulence factors by pathogens into host cells to modulate cellular functions. For example, *Porphyromonas gingivalis* OMVs contain its major virulence factors gingipains, which can degrade cytokines to inhibit host inflammatory response [46]; *Campylobacter jejuni* OMVs carry 16 N-linked glycoproteins, which can be delivered into host cells to provoke an immune response [47]. Whether *E.-coli*-K1-derived OMVs carry virulence factors and their role in regulating the immune functions of microglia are currently unknown and require further investigation.

Astrocytes are the most abundant cell type in the CNS and have been implicated in defending the CNS against infections to maintain neuronal health and function [48]. Astrocytes have recently been reported to display intricate interactions with microglia to elicit inflammatory reactions in the CNS during injury and disease, and many studies have highlighted the importance of microglia-astrocyte crosstalk [38]. On one hand, astrocyte activation has been reported to be dependent on microglial activation because microglia are more rapidly activated in response to bacterial challenge [16]. Cytokines (including TNF-α) and many other secreted mediators that are released by activated microglia can activate astrocytes, converting them into their pro-inflammatory phenotype [49]. On the other hand, astrocytes can balance the inflammatory levels of microglia, for example, by releasing orosomucoid-2 to inhibit pro-inflammatory microglia to avoid exaggerated inflammatory responses [50]. Our work shows that during *E. coli* K1 infection of the brain, astrocytes are activated by microglia-derived TNF-α, which then secrete chemokine CXCL1 to promote neutrophil recruitment, which is required to control bacterial infection. Astrocytes can produce CXCL1 [30], but the precise role of astrocyte-derived CXCL1 during *E. coli* K1 CNS infection is unclear. Our results indicate that astrocyte-derived CXCL1 is essential for neutrophil recruitment to the CNS to defend against *E. coli* K1 infection. Although microglia also have the ability to produce CXCL1 [29], microglia-derived CXCL1 levels might be significantly lower than those produced by astrocytes, as evidenced by the immunofluorescence assays in this study. Thus, our results highlight the importance of microglia–astrocyte crosstalk in controlling bacterial infections.

Our results indicate that there must be an effective anti-inflammatory response in the CNS to avoid tissue damage induced by pro-inflammatory cytokines or neutrophils, since both the production of TNF-α and neutrophil recruitment decreased 24 h post-infection. In addition, we did not observe any noticeable behavioral abnormalities in *E.-coli*-K1-infected mice 3 days post-infection. The balance between inflammatory and anti-inflammatory responses is important for the maintenance of CNS homeostasis and neuronal function, but the underlying mechanisms are unknown, especially because astrocytes are known to inhibit the inflammation caused by microglia [50]. Whether the crosstalk between microglia and astrocytes contributes to this balance during *E. coli* K1 CNS infection requires further study.

A limitation of this study is the relatively small number of study subjects (4 or 6 mice) per group. A larger sample size is necessary in future studies to fully understand the pathogenesis of *E.-coli*-K1-induced meningitis and how this bacteria interacts with the host.

In summary, our study demonstrated the recruitment of neutrophils to the CNS during *E. coli* K1 infection and identified its underlying mechanisms. We present a complex network of host and bacterial factors and highlight the collaborative action of microglia and astrocytes to control CNS *E. coli* K1 infection. The involvement of other CNS resident cells, such as endothelial cells, in the recruitment of neutrophils and their role in defense against *E. coli* K1 infection require further study.

## 4. Materials and Methods

### 4.1. Bacterial Culture Conditions

*Escherichia coli* K1 strain RS218 (serotype O18:K1:H7) was isolated from the cerebrospinal fluid of a newborn infant with meningitis. The Δ*neuDB*, Δ*ypjA*, Δ*yrbE*, and Δ*msbB* mutant strains were constructed by homologous recombination. All the bacterial strains were cultured in Luria–Bertani (LB) broth at 37 °C with shaking (200 rpm), and bacterial growth was detected using spectrophotometer at 600 nm (OD_600_).

### 4.2. Isolation of Bacterial OMVs

To prepare OMVs, bacterial cells were cultured in LB broth at 37 °C with shaking until the mid-exponential phase (OD_600_~0.6), and cultures were pelleted at 4000× *g* for 5 min at 4 °C. The supernatants were filtered through a 0.45 μm pore size filter, and OMVs were pelleted by ultracentrifugation at 150,000× *g* for 3 h at 4 °C. After removing the supernatant, the pellet was washed twice with PBS. The OMVs were resuspended in 500 μL PBS and stored at −80 °C.

### 4.3. Isolation of Microglia and Astrocytes from Mouse Brain

Primary microglia and astrocytes were prepared from 3- to 5-day-old mice brains using an enzyme-based method [51,52]. Briefly, brains were dissected, minced, and dissociated with 0.25% trypsin supplemented with EDTA for 15 min at 37 °C. The reaction was stopped by the addition of fetal bovine serum (FBS), and the samples were filtered consecutively through strainers with a pore size of 70 μm. Cells were resuspended in Dulbecco’s modified Eagle’s medium (DMEM) supplemented with 10% FBS and 100 U/mL penicillin/streptomycin and plated in cell culture flasks. After 10 days, the flasks were gently shaken for 3 h, and the medium was harvested and centrifuged for 10 min at 1000× *g* to collect microglial cells. For basal astrocyte isolation, 20 mL fresh medium was added to the cell culture flasks and shaken at 240 rpm for 6 h to remove residual microglia.

### 4.4. Cell Culture and Stimulations

BV2 cells (an immortalized murine microglial cell line) were cultured in DMEM medium supplemented with 10% FBS and 100 U/mL of penicillin/streptomycin at 37 ℃ under 5% CO_2_. For the anti-phagocytic ability measurement of *E. coli* K1, BV2 cells were seeded in 12-well plates at a density of 2 × 10^5^ cells/well and cultured with GFP-labeled bacteria at an MOI of 5 for 1 h. The cells were washed thrice with PBS to remove extracellular bacteria and dissociated with trypsin. The reaction was stopped with FBS, and the cells were pelleted after centrifugation at 1000× *g* for 5 min. The cells were resuspended in PBS and stained with fluorophore-conjugated antibodies in the presence of 0.5% BSA for 30 min at 4 °C. FACS was performed to determine the percentage of cells containing bacteria.

Before stimulating microglia with OMVs or bacteria, isolated primary microglia were cultured in astrocyte-conditioned medium (ACM). For preparation of ACM, isolated astrocytes were cultured in DMEM medium supplemented with 10% FBS and 100 U/mL penicillin/streptomycin for 48 h. ACM was collected and filtered through a 0.45 μm pore size filter. Enriched microglia were plated in 24-well dishes at a density of 5 × 10^4^ cells/well and maintained in ACM at 37 °C and 5% CO_2_ for 24 h. After this, ACM was removed, and primary microglia were stimulated with OMVs diluted in DMEM for 2 h. Supernatant was collected and stored at −80 °C before analysis. For bacterial stimulation, primary microglia were plated in the lower chamber of a 0.45 μm transwell plate at a density of 5 × 10^4^ cells/well, and bacteria that were collected and resuspended in DMEM medium were added into the upper chamber at an MOI of 1 for 2 h. Cell supernatants were collected and stored at −80 °C before analysis.

Astrocytes were plated into 12-well plates at a density of 5 × 10^4^ cells/well, and the cell supernatants were replaced with new medium to remove microglia before stimulation with TNF-α, microglia-conditioned medium (MCM), or TNF receptor inhibitor. Cell supernatant was collected and stored at −80 °C before analysis.

Cytokines and chemokines were measured using ELISA (R&D Systems) following the manufacturer’s instructions.

### 4.5. Isolation of CNS Neutrophils and Analysis of Their Recruitment

Mice were injected with (via intravenous injection) 2.5 × 10^6^ CFU of *E. coli* K1 WT strain dissolved in 100 μL PBS or with 100 μL PBS (uninfected control). At indicated time points post-infection (4, 12, 24, and 48 h), uninfected and infected mice were euthanized and perfused with normal saline. The brains were harvested, and neutrophils were isolated from brain single-cell suspensions using a discontinuous Percoll gradient [37,53]. Briefly, the excised brains were ground using a rubber-attached disposable syringe plunger through a steel mesh to prepare single-cell suspensions. The suspension was added to 7 mL of FACS buffer (HBSS + 0.5% BSA) and 3 mL of 90% Percoll in PBS was added. Then, the suspension was underlaid with 1 mL of 70% Percoll and centrifuged at 2450 rpm for 20 min at 4 °C. The cells that accumulated at the interface of the gradient were collected, washed three times with FACS buffer, and resuspended in PBS. Cells were then stained with fluorophore-conjugated antibodies in the presence of 0.5% BSA for 30 min at 4 °C. The anti-mouse antibodies used in this study were CD45, CD11b, Ly-6G, and Ly-6C (eBiosciences). Samples were washed in FACS buffer, and FACS was performed. Finally, the data were analyzed using the FlowJo software (v10.8, BD Biosciences, San Jose, CA, USA).

### 4.6. Cytokine and Chemokine Quantification in Brain Tissue

Mice were injected with 2.5 × 10^6^ CFU of *E. coli* K1 WT strain dissolved in 100 μL PBS. Infected brains were dissected at the designated point in time and homogenized in 1.5 mL PBS with a protease inhibitor cocktail. Samples were centrifuged to remove debris and stored at −80 °C before analysis. Cytokines and chemokines were measured using ELISA (R&D Systems, Minneapolis, MN, USA) following the manufacturer’s instructions.

### 4.7. Quantitative RT-PCR

Mice were injected with (via intravenous injection) 2.5 × 10^6^ CFU of *E. coli* K1 WT strain dissolved in 100 μL PBS or with 100 μL PBS. At indicated time points post-infection, uninfected and infected mice were euthanized. RNA was extracted from *E.-coli*-K1- or PBS-treated brains using TRIzol (Invitrogen, Waltham, MA, USA) and an RNeasy kit (Qiagen, Germantown, MD, USA) according to the manufacturer’s instructions. Purified RNA was used as a template for cDNA generation using the PrimeScript™ RT Master Mix (Takara, shiga, Japan) with oligo(dT) and random primers. Quantitative real-time PCR (RT-PCR) was performed using Fast SYBR Green (Thermo Fisher Scientific) and 10 nM primers designed using PrimerDesign (Thermo Fisher Scientific, Waltham, MA, USA). The expression of each gene was calculated relative to GAPDH using the ΔΔCT method.

### 4.8. Fluorescence Microscopy of OMVs

To stain OMVs, BV2 cells were grown on coverslips and infected with *E. coli* K1 for 2 h. After stimulation, the cells were fixed in 4% paraformaldehyde, permeabilized with 0.1% Triton-X, and blocked with 5% goat serum for 2 h prior to staining overnight at 4 °C with antibodies against LPS, followed by staining with fluorescently labeled secondary antibodies for 1 h and 1 μg/mL Hoechst-33258 for 20 min. Subsequently, the cells were sealed with a fluorescence quencher and visualized under a Zeiss LSM 800 microscope.

### 4.9. Transmission Electron Microscopy (TEM) of OMVs

BV2 cells were grown on coverslips and infected with *E. coli* K1 for 2 h. After stimulation, the cells and bacteria were collected and fixed with 4% paraformaldehyde at room temperature for 1 h, followed by further fixation at 4 °C. Samples were post-fixed in 1% tannic acid for 1 h, followed by 1% osmium tetroxide treatment for 1 h, and staining with 1% uranyl acetate. Subsequently, samples were dehydrated using a graded ethanol series. Samples were infiltrated into propylene oxide and polymerized at 60 °C for 48 h. The resulting blocks were sectioned at 70 nm on 300-mesh copper grids and imaged using a transmission electron microscope at 100 kV.

### 4.10. Immunohistochemistry of Brain Sections

Mice (3-week-old) were infected with *E. coli* K1 via tail vein. After 12 h, the mice were perfused and their brains were fixed in 3.7% formaldehyde for 48 h, followed by a graded sucrose dehydration. Tissue sections (20 μm) were prepared on a Leica Cm1950 cryostat platform. The slices were fixed in 4% paraformaldehyde, permeabilized with 0.1% Triton-X, and blocked with 5% BSA for 2 h prior to staining overnight at 4 °C with antibodies against CXCL1, GFAP, and Iba1, followed by staining with fluorescently labeled secondary antibodies for 1 h and 1 μg/mL Hoechst-33258 for 20 min. Slices were visualized using a Zeiss LSM800 confocal microscope. Each experiment was repeated at least thrice.

### 4.11. In Vitro Transmigration Assays

Bone marrow neutrophils from 6-week-old mice were isolated using the EasySep™ Mouse Neutrophil Enrichment Kit (StemCell, Vancouver, BC, Canada). Transwell filters (5 μm, Corning Costar, Kennebunk, ME, USA) were plated with confluent bEnd.3 cells and cultured overnight. Isolated neutrophils were pretreated with CXCR2 antibodies or isotype antibodies for 3 h and added to the upper chamber in serum-free RPMI medium. Transmigration was induced by adding the cellular supernatant derived from the whole *E.-coli*-K1-infected brain. Transmigrated cells were counted at 3 h and expressed as a percentage of the total cells added [35].

### 4.12. Statistical Analyses

All experimental data were analyzed using GraphPad software. Student’s unpaired *t*-test, or Mann–Whitney U test was performed according to the requirements of the data. In all cases, *p* values < 0.05 were considered significant. The in vitro experiments were conducted in triplicate. The animal experiments were conducted in duplicate, and the combined data from the two experiments were used for statistical analysis.

## Figures and Tables

**Figure 1 ijms-23-06540-f001:**
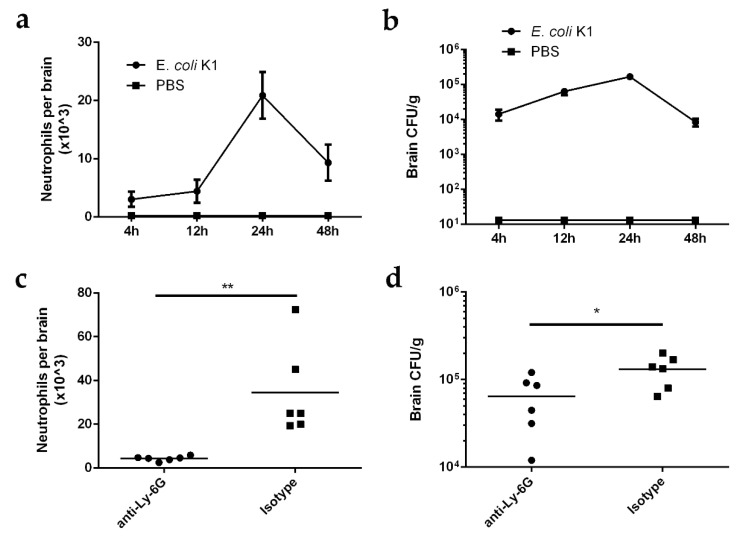
*Escherichia coli* K1 induces neutrophil recruitment into CNS of infected mice. (**a**,**b**) Neutrophil counts (**a**) and bacterial loads (**b**) in the mouse brain 4, 12, 24, and 48 h after intravenous injection with *E. coli* K1 or PBS. Data are presented as mean ± SD, *n* = 6 mice per group. (**c**,**d**) Neutrophil counts (**c**) and bacterial loads (**d**) in the Ly-6G-antibody-treated or isotype-treated (negative control) mouse brain 24 h after intravenous injection with *E. coli* K1. *n* = 6 mice per group. *p* values were determined using Mann–Whitney U test. * *p <* 0.05; ** *p <* 0.01.

**Figure 2 ijms-23-06540-f002:**
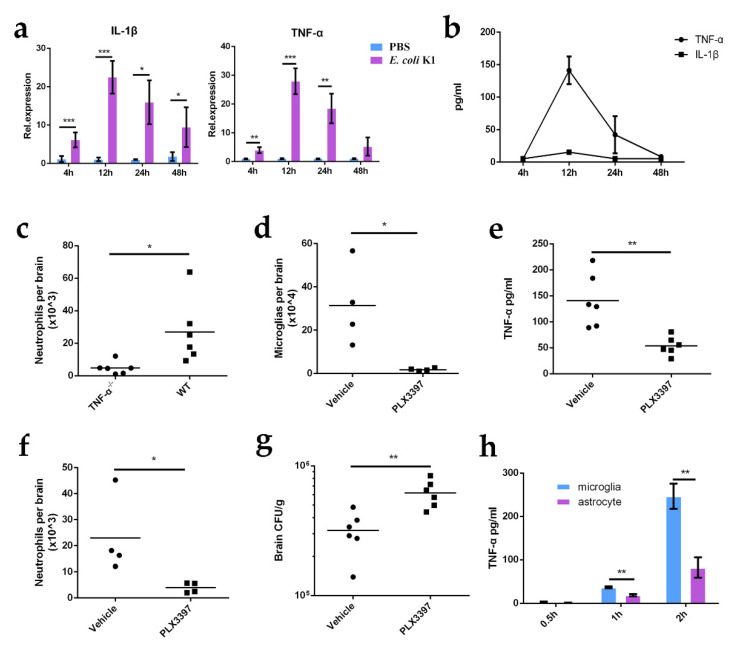
*E.-coli*-K1-induced neutrophil recruitment requires TNF-α, microglia-derived pro-inflammatory cytokine. (**a**) Real-time quantitative PCR analysis of TNF-α and IL-1β mRNA levels in uninfected mouse brains and those infected with *E. coli* K1 for 4, 12, 24, and 48 h (*n* = 6 mice per group). (**b**) ELISA analysis of TNF-α and IL-1β protein levels in *E.-coli*-K1-infected brain 4, 12, 24, and 48 h post-infection (*n* = 6 mice per group). (**c**) Neutrophil counts in the brains of wild-type (WT) mice and TNF-α-deficient mice infected with *E. coli* K1 for 24 h (*n* = 6 mice per group). (**d**–**g**) Mice were pretreated with PLX3397 or vehicle (negative control) for 2 weeks and then infected with *E. coli* K1 for 24 h. Thereafter, the microglia count ((**d**), *n* = 4 mice per group), TNF-α protein levels ((**e**), *n* = 6 mice per group), neutrophil counts ((**f**), *n* = 4 mice per group), and bacterial loads ((**g**), *n* = 6 mice per group) in the mouse brain were analyzed. (**h**) Primary microglia and astrocytes were stimulated with *E. coli* K1 for 0.5, 1, 2 h, and the production of TNF-α was assessed using ELISA. Data are presented as mean ± SD, *n* = 3 independent experiments. *p* values were determined using Student’s *t*-test (**a**,**h**) or Mann–Whitney U test (**c**–**g**). * *p <* 0.05; ** *p <* 0.01; *** *p <* 0.001.

**Figure 3 ijms-23-06540-f003:**
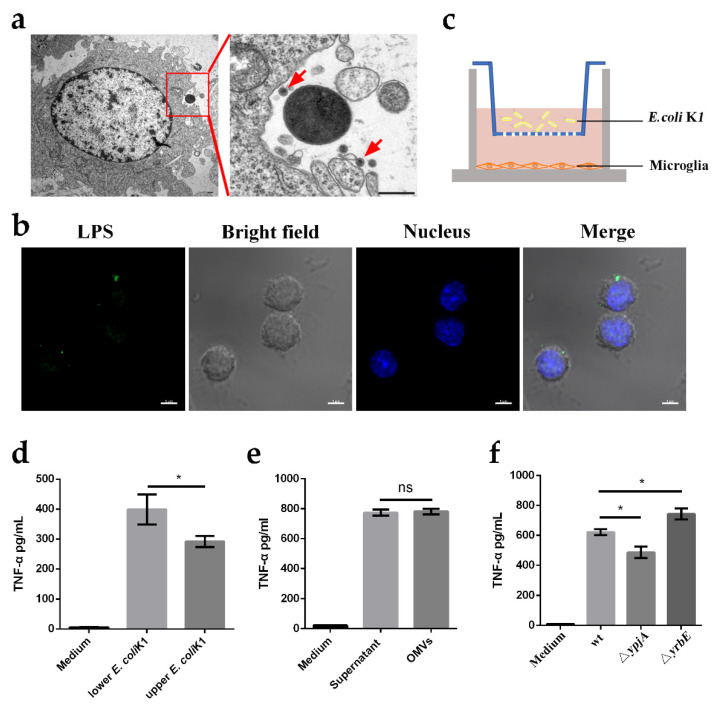
Microglia recognize *E.-coli*-K1-derived outer membrane vesicles (OMVs) to release TNF-α. (**a**,**b**) Transmission electron microscopy (**a**) and confocal laser microscopy (**b**) were utilized to observe the secretion of OMVs by *E. coli* K1 when the bacteria were cultured with primary microglia. Images are representative of three independent experiments. High magnification image of red boxed area is shown on the right panel and red arrows indicate the observed OMV (**a**). Scale bars, 500 nm (**a**) and 20 μm (**b**). (**c**) Experiment scheme to determine the inflammatory activation of microglia by bacteria-secreted factors using Transwell plate. (**d**) ELISA analysis of TNF-α protein levels of BV2 microglia cell line incubated with wild-type *E. coli* K1 in the lower wells or upper wells. (**e**) ELISA analysis of TNF-α protein levels in microglia that were stimulated with bacterial supernatant or purified OMVs. (**f**) ELISA analysis of TNF-α protein levels in microglia that were cultured with wild-type *E. coli* K1, Δ*ypjA*, or Δ*yrbE*. Data are presented as mean ± SD, *n* = 3 independent experiments. *p* values were determined using Student’s *t*-test (**d**–**f**). * *p <* 0.05; ns, not significant.

**Figure 4 ijms-23-06540-f004:**
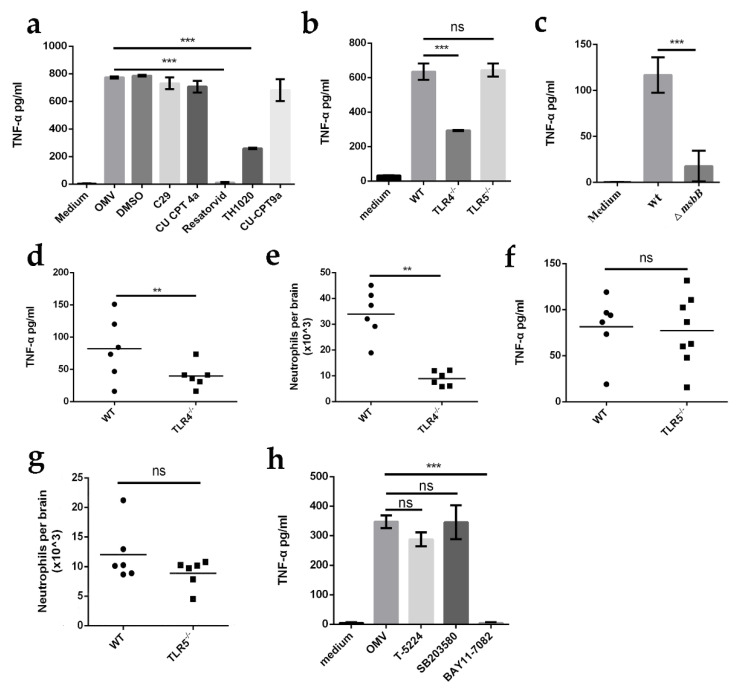
Microglia TLR4 recognizes LPS of the *E.-coli*-K1-derived OMV to release TNF-α. (**a**) ELISA of TNF-α protein levels in BV2 microglial cell line incubated with *E.-coli*-K1-derived OMVs for 2 h in the presence of TLR2 inhibitor C29, TLR3 inhibitor CU CPT4a, TLR4 inhibitor Resatorvid, TLR5 inhibitor TH1020, or TLR8 inhibitor CU-CPT9a. (**b**) ELISA of TNF-α protein levels in primary microglia derived from wild-type mice (WT), TLR4 deficient mice (*TLR4*^−/−^), and TLR5 deficient mice (*TLR5*^−/−^) after incubation with *E.-coli*-K1-derived OMVs for 2 h. (**c**) ELISA analysis of TNF-α protein levels in primary microglia that were incubated with *E. coli*-K1-wild-type- (wt) or Δ*msbB*-derived OMVs for 2 h. (**d**–**g**) *TLR4* or *TLR5* deficient mice were infected wild-type with *E. coli* K1 for 12 h. Thereafter, the TNF-α protein levels ((**d**,**f**); *n* = 6 mice per group) and neutrophil counts ((**e**,**g**); *n* = 6 mice per group) in the mouse brain were analyzed. (**h**) ELISA analysis of TNF-α protein levels in primary microglia that were pretreated with BAY11-7082 (NF-κB inhibitor), T-5224 (c-Fos/AP-1 inhibitor), or SB203580 (p38 MAPK inhibitor) for 2 h, and then incubated with *E.-coli*-K1-derived OMVs for 2 h. Data are presented as mean ± SD, *n* = 3 independent experiments (**a**–**c**,**h**). *p* values were determined using Student’s *t*-test (**a**–**c**,**h**) or Mann–Whitney U test (**d**–**g**). ** *p <* 0.01; *** *p <* 0.001; ns, not significant.

**Figure 5 ijms-23-06540-f005:**
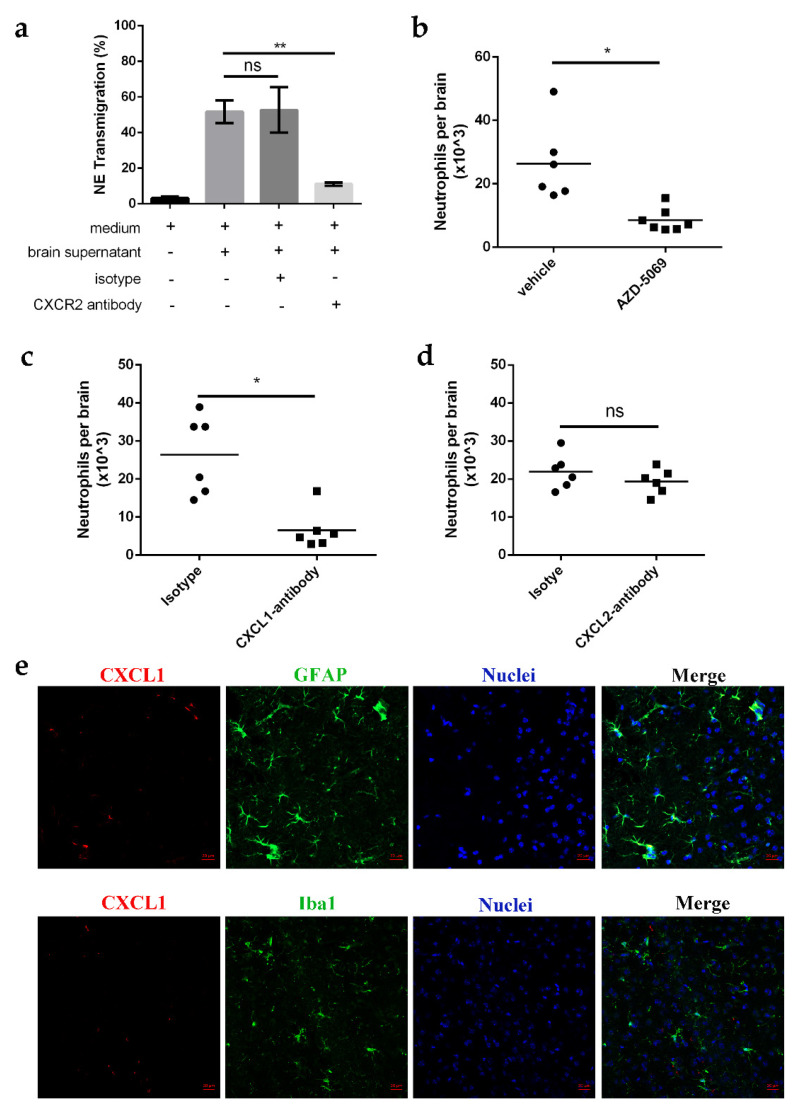
Astrocyte-derived chemokine CXCL1 is required for neutrophil recruitment to CNS during *E. coli* K1 infection. (**a**) Primary neutrophils isolated from bone marrow of 6-week-old mice were treated with either CXCR2 antibody or isotype control before inducing neutrophil transmigration. Transmigrated cells were counted 3 h after infection with *E. coli* K1. Data are presented as mean ± SD, *n* = 3 independent experiments. (**b**) Neutrophil counts in the brains of mice that were pretreated with AZD-5069 (a CXCR2 inhibitor) or vehicle (negative control) and then infected with *E. coli* K1 for 24 h (*n* = 6 mice for vehicle; *n* = 6 mice for AZD-5069). (**c**,**d**) Neutrophil counts in the brains of mice that were co-injected with *E. coli* K1 and CXCL1 (**c**) and CXCL2 (**d**) antibody for 12 h (*n* = 6 mice per group). (**e**) Confocal laser microscopy analysis showed that CXCL1 co-localized with astrocytes (GFAP^+^) but not microglia (Iba1^+^). Scale bars, 20 μm. Images are representative of three independent experiments. *p* values were determined using Student’s *t*-test (**a**) or Mann–Whitney U test (**b**–**d**). * *p <* 0.05; ** *p <* 0.01; ns, not significant.

**Figure 6 ijms-23-06540-f006:**
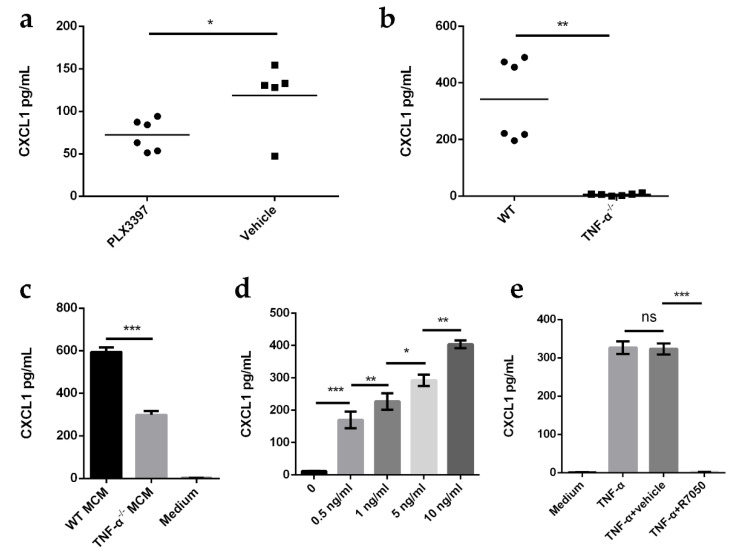
TNF-α produced from microglia promotes CXCL1 release by astrocytes. (**a**) Mice were pretreated with PLX3397 or vehicle (negative control) for 2 weeks, and then infected with *E. coli* K1 for 12 h. CXCL1 protein levels in the mouse brain were analyzed by ELISA (*n* = 5 mice for vehicle; *n* = 6 mice for PLX3397). (**b**) ELISA analysis of CXCL1 protein levels in the brains of wild-type (WT) mice and TNF-α-deficient mice infected with *E. coli* K1 for 12 h (*n* = 6 mice per group). (**c**) ELISA analysis of CXCL1 protein levels in astrocytes cultured with wild-type (WT) or *TNF-α*^−/−^ microglia-conditioned medium (MCM) for 2 h. (**d**) ELISA analysis of CXCL1 protein levels in astrocytes treated with different concentrations of TNF-α. (**e**) Primary astrocytes were pretreated with TNF-α receptor antagonist R-7050 or vehicle (negative control) for 1 h and then treated with TNF-α for 2 h. CXCL1 protein levels in astrocytes were analyzed by ELISA. Data are presented as mean ± SD, *n* = 3 independent experiments (**c**–**e**). *p* values were determined using Mann–Whitney U test (**a**,**b**) or Student’s *t*-test (**c**–**e**). * *p <* 0.05; ** *p <* 0.01; *** *p* < 0.001; ns, not significant.

## Data Availability

Not applicable.

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
