# Peer review of "Collaborative Action of Microglia and Astrocytes Mediates Neutrophil Recruitment to the CNS to Defend against Escherichia coli K1 Infection"

_ijms, 2022, doi:10.3390/ijms23126540_

Round 1

Reviewer 1 Report

This paper provides evidence supporting a functional role for neutrophils E. coli infections highlighting the differential regulation of neutrophil recruitment in the brain. 

It is known that TNF-α induces an NF-κB-dependent CXCL1 increase in cultured astrocytes. Taking into account these findings the authors hypothesized that microglia-secreted TNF-α promotes the release of CXCL1 from astrocytes in the E. coli K1-infected brain. Indeed this hypothesis proved to be valid: using  TNF-α antagonist PLX3397 and deficient mice  E. coli K1 infection failed to produce CXCL1 in the brain compared to that in the respective controls. These findings indicate that microglia-derived TNF-α plays a critical role in the production of CXCL1. This result was further confirmed in vitro experiments. This is an original finding.

Neutrophils plays a critical role in defense against E. coli K1 infection; however, the mechanism underlying this recruitment remains unclear. In this study the authors reported that microglia and astrocytes are activated in response to stimulation by E. coli. This study also reveals that (E. coli) outer membrane vesicles are sufficient to induce microglial activation producing TNF-α and subsequent CXCL1 production. The production of inflammatory cytokines is subjected to noradrenaline via β-adrenoceptor activation (J of Neuroimmunology 61:123-131, 1995) resulting in inhibition. Have you studied the role of β-adrenoceptor in your findings? 

Author Response

This paper provides evidence supporting a functional role for neutrophils E. coli infections highlighting the differential regulation of neutrophil recruitment in the brain. 

It is known that TNF-α induces an NF-κB-dependent CXCL1 increase in cultured astrocytes. Taking into account these findings the authors hypothesized that microglia-secreted TNF-α promotes the release of CXCL1 from astrocytes in the E. coli K1-infected brain. Indeed this hypothesis proved to be valid: using  TNF-α antagonist PLX3397 and deficient mice  E. coli K1 infection failed to produce CXCL1 in the brain compared to that in the respective controls. These findings indicate that microglia-derived TNF-α plays a critical role in the production of CXCL1. This result was further confirmed in vitro experiments. This is an original finding.

Response :  Thank you for your patient and thoughtful reading as well as the comments about our manuscript.

Point 1: Neutrophils plays a critical role in defense against E. coli K1 infection; however, the mechanism underlying this recruitment remains unclear. In this study the authors reported that microglia and astrocytes are activated in response to stimulation by E. coli. This study also reveals that (E. coli) outer membrane vesicles are sufficient to induce microglial activation producing TNF-α and subsequent CXCL1 production. The production of inflammatory cytokines is subjected to noradrenaline via β-adrenoceptor activation (J of Neuroimmunology 61:123-131, 1995) resulting in inhibition. Have you studied the role of β-adrenoceptor in your findings? 

Response : Thank you for the comment. The role of β-adrenoceptor has not been studied in this work and will be investigated in our future work.

Because β-adrenoceptor can regulate the production of inflammatory factors, it is possible that β-adrenoceptor might be used as a potential therapeutic target to treat the damage caused by E. coli K1-induced excessive inflammation.

Reviewer 2 Report

Dear Authors,

thank you very much for your nice-written work. Please pay attention to the following questions and queries of mine, after reviewing your manuscript:

1.      Line 99-100: you induced a bacterial meningitis to your study subjects after intravenously injected an E. coli K1 strain, gained from an infant with meningitis, to a 3-week-old mice. Do you believe that the blood–brain barrier (BBB) could have influenced the bacterial presence in the CSF of your study subjects and through this the overall in vivo results of your study?

2.      Line 134: you mention that your infected study subjects were euthanized 12 hpi. This means that all the cellular and extracellular molecules of the study brains were measured at this particular time point. Please explain how did you gain cellular and extracellular material in order to conduct your measurements earlier (4hpi) and later (24, 48 hpi) than 12 hpi. For example, according to the Line 142: both TNF-α and IL-1β were measured in extracellular supernatant on 4, 12 and 24 hpi.

3.      Line 145: what do you mean by referring to the lack of mature IL-1β? What is exactly a mature IL-1β molecule and why was not present in extracellular supernatant despite its intracellular mRNA production?

4.      Line 399-400: please correct the grammar error: “which is activates” as “which activates” or “which is activating”.

5.      Please discuss the small number of study subjects (4 or 6 mice) per group as a limitation of your study.

With Best Regards

Author Response

Point 1: Line 99-100: you induced a bacterial meningitis to your study subjects after intravenously injected an E. coli K1 strain, gained from an infant with meningitis, to a 3-week-old mice. Do you believe that the blood–brain barrier (BBB) could have influenced the bacterial presence in the CSF of your study subjects and through this the overall in vivo results of your study?

Response 1: Thank you for the comment. It has been reported that E. coli K1 can cross the BBB by transcellular penetration (Infect Immun. 1999, 67(11): 5775–5783). With similar levels of bacteraemia, E. coli K1 can penetrate into the CNS of infant and adult animals at similar levels (J. Clin. Invest. 1992, 90: 897–905). These results indicate that BBB might has little influence on the bacterial presence in the CSF.

Moreover, it has been recognized that the bacterial loads in the blood can influence the bacterial presence in the CSF, and a high degree of bacteraemia is necessary for the development of meningitis by E. coli K1 (Infect Immun. 1996, 64:154–160; Nature Reviews Microbiology, 2008, 6(8): 625-634).

Point 2: Line 134: you mention that your infected study subjects were euthanized 12 hpi. This means that all the cellular and extracellular molecules of the study brains were measured at this particular time point. Please explain how did you gain cellular and extracellular material in order to conduct your measurements earlier (4hpi) and later (24, 48 hpi) than 12 hpi. For example, according to the Line 142: both TNF-α and IL-1β were measured in extracellular supernatant on 4, 12 and 24 hpi.

Response 2: Thank you for the comment. In this work, we have also tested the mRNA levels of TNF-α and IL-1β in E. coli K1-infected or mock-infected mouse brain at 4, 12 and 48 hpi, in addition to that of 12 hpi. The results showed that the mRNA levels of TNF-α and IL-1β were increased in the brains of K1-infected mice, at all time points tested, and reached their peak at 12 hpi. In the original manuscript, we only presented the results of 12 hpi. To avoid confusion, We have now presented all the qRT-PCR results in the revised Figure 2a, and revised the manuscript accordingly (line 137 and line 163).

Point 3: Line 145: what do you mean by referring to the lack of mature IL-1β? What is exactly a mature IL-1β molecule and why was not present in extracellular supernatant despite its intracellular mRNA production?

Response 3: Thank you for the comment. Cytokines are mostly secreted into the extracellular matrix to mediate immune response. In order to explore whether IL-1β is involved in neutrophil recruitment induced by E. coli K1, we measured the protein levels of IL-1β in the extracellular supernatant derived from K1-infected mouse brains. However, IL-1β protein was not detectable at any of the time points tested, although its mRNA level was significantly increased. This may be correlated with the expression, maturation and secretion of IL-1β.  IL-1β is expressed and synthesized in microglia as an inactive precursor protein that requires cleavage by caspase-1 to be transformed into its mature form, and only the mature IL-1β can be secreted into extracellular space (Cytokine Growth Factor Rev. 2011, 22(4): 189–195; J Exp Med 2020, 217 (1): e20190314.). The reason why IL-1β was not detected in extracellular supernatant is probably due to the absence of mature IL-1β. To avoid confusion, we now added the information in the revised manuscript (lines 148-151).

Point 4: Line 399-400: please correct the grammar error: “which is activates” as “which activates” or “which is activating”.

Response 4: “which is activates” is now changed to “which activates” in the revised manuscript (line 410).

Point 5: Please discuss the small number of study subjects (4 or 6 mice) per group as a limitation of your study.

Response 5: As suggested, we have discussed this point in the revised manuscript (line 490-492).

Reviewer 3 Report

In this manuscript Liu et al. report that E. coli K1, a leading cause of neonatal meningitis, induces the release of the proinflammatory cytokine TNFα by microglia. In turn, TNFα activates astrocytes resulting in the production of the chemokine CXCL1 which triggers the migration of neutrophils within the central nervous system. This is a very interesting observation that highlight the collaborative action between the different actors of cerebral immunity (microglia, astrocytes and peripheral immune cells). The manuscript is well written and very clear.

I have few questions and minors’ comments regarding the manuscript

It is not clear if neutrophil infiltration occurs in the meninges or also in the brain parenchyma? According to Fig. S1 it seems that neutrophils infiltrate mainly the meninges. One sentence should be added in the manuscript for clarify this point.

The bacterial inoculum used to infect mice is not mentioned.

Which time point has been used for Fig. 1C and 1D?

Fig. 3A and B: please indicate what the scale bars correspond to. Red arrows (3A) showing OMV must be stated in the legend. A higher magnification for Fig. 3A and B would allow a better visualization of OMV.

Fig. 5A, B, C, D are too small.

Fig. 5E: Please indicate what the scale bars correspond to.

Author Response

Point 1: It is not clear if neutrophil infiltration occurs in the meninges or also in the brain parenchyma? According to Fig. S1 it seems that neutrophils infiltrate mainly the meninges. One sentence should be added in the manuscript for clarify this point.

Response 1: Thank you for the comment. Through confocal microscopy observation, we found that neutrophils infiltrate mainly the meninges. We have added this information in the revised manuscript (line 104-105).

Point 2: The bacterial inoculum used to infect mice is not mentioned.

Response 2: For the tail vein injection model, mice were injected with 2.5×106 CFU bacteria dissolved in 100 μl phosphate-buffered saline (PBS) or with 100 μl PBS (uninfected control). We have added this information in the Materials and Methods section of the revised manuscript (lines 554-556, 570 and 576-578).

Point 3: Which time point has been used for Fig. 1C and 1D?

Response 3: Thank you for pointing this out. The time point used for Fig. 1C and 1D is at 24 h after intravenous injection with E. coli K1, and we have updated this information in the legend of Figure. 1 (line127).

Point 4: Fig. 3A and B: please indicate what the scale bars correspond to. Red arrows (3A) showing OMV must be stated in the legend. A higher magnification for Fig. 3A and B would allow a better visualization of OMV.

Response 4: Thank you for the comment. We have added the scale bars of Figure. 3a and b in the legend (line 240), and red arrows showing OMV was also described in the legend (line 239). A higher magnification for Fig 3a is now provided. However, we have not deal with Figure. 3b, due to that the enlarge will significantly reduce the clarity of this picture.

Point 5: Fig. 5A, B, C, D are too small.

Response 5: We have adjusted Figure 5, to allow a better visualization of Figure. 5a, b, c, d.

Point 6: Fig. 5E: Please indicate what the scale bars correspond to.

Response 6: The scale bar of each picture is 20 μm, we have added this information to the legend of Figure. 5 (line 362).